# Multiplex Assay for Rapid Detection and Analysis of Nucleic Acid Using Barcode Receptor Encoded Particle (BREP)

**DOI:** 10.3390/biomedicines10123246

**Published:** 2022-12-13

**Authors:** Semyung Jung, Ki Wan Bong, Wonhwi Na

**Affiliations:** 1Department of Chemical and Biological Engineering, Korea University, Seoul 02841, Republic of Korea; 2Engineering Research Center for Biofluid Biopsy, Seoul 02841, Republic of Korea

**Keywords:** molecular diagnosis, hydrogel microparticle, DNA, multiplex

## Abstract

Several multiplex nucleic acid assay platforms have been developed in response to the increasing importance of nucleic acid analysis, but these assays should be optimized as per the requirements of point-of-care for clinical diagnosis. To achieve rapid and accurate detection, involving a simple procedure, we propose a new concept in the field of nucleic acid multiplex assay platforms using hydrogel microparticles, called barcode receptor-encoded particles (BREPs). The BREP assay detects multiple targets in a single reaction with a single fluorophore by analyzing graphically encoded hydrogel particles. By introducing sets of artificially synthesized barcode receptor and barcode probes, the BREP assay is easily applicable in multiplexing any genetic target; sets of barcode receptors and barcode probes should be designed delicately for universal application. The performance of the BREP assay was successfully verified in a multiplex assay for the identification of different malaria species with high sensitivity, wide dynamic range, fast detection time, and multiplexibility.

## 1. Introduction

In recent years, significant progress has been made in the development of biomolecule-based diagnostics. Information regarding biomolecules is essential for the development of clinical diagnostics, drug discovery, and basic science research [1,2]. Moreover, the sudden emergence of the coronavirus disease in 2019 (COVID-19), caused by the severe acute respiratory syndrome coronavirus 2 (SARS-CoV-2), resulted in a severe public health crisis, in which rapid and accurate screening of patients became mandatory [3,4]. Furthermore, because the emergence of SARS-CoV-2 variants, including alpha, beta, gamma, delta, and lambda variants, has increased transmission and resistance to vaccines or immunity, timely detection and accurate identification of such variants is essential [5,6]. Therefore, efficient detection and tracking methods are required [7]. Nucleic acid-based assays can fulfill clinical demands owing to their high specificity and sensitivity [8]. At present, nucleic acid-based assays are required to detect multiple biomarkers simultaneously in a single reaction, reducing time, cost, and labor. The amplification of multiple nucleic acids enables the acquisition of diagnostic information, e.g., identification of infectious agents, genetic variations, mutant genes, or single nucleotide polymorphism (SNP) related to particular diseases, even though the amount of sample that can be acquired is limited or the concentration of analytes in the sample is low; this is not possible in the case of single-target serial testing. [9,10,11]. Current platforms used for multiplexing can be divided into two broad classes: (1) planar microarrays and (2) suspension arrays [2,12,13].

DNA microarrays utilize the fundamental properties of nucleic acids to detect complementary sequences by DNA–DNA hybridization [14]. Target-specific receptors are immobilized at a “spot” on a solid surface, enabling positional encoding of large numbers of targets. The labeled test sample in the solution phase is introduced on the solid surface and then allowed to hybridize with the immobilized reporters by diffusion [15,16,17]. Although this method can accommodate high-level multiplexing, it suffers from low throughput due to slow diffusion to the solid surface and inflexible chip fabrication, which make it difficult to meet clinical demands [9,11,18].

Suspension (bead-based) arrays use particles conjugated to probes specific to each target nucleotide sequence, and these particles, considered like each spot on a microarray, are identifiable by spectrometric, electronic, graphical, or physical encoding. In this assay, the test samples are reacted with a suspension of these particles in the solution phase. Suspension arrays have a significant advantage over planar microarrays, as they provide faster reaction kinetics due to the reaction being performed in an aqueous solution and due to the increased surface area of three-dimensional space, unlike microarrays with a two-dimensional planar surface [18,19,20]. When combined with powerful applications, these particles could be used for screening thousands of biomolecules in the laboratory, thus replacing conventional analysis methods, which are time-consuming, complex, and expensive [11]. Extensive efforts are being made to develop suspension array platforms with improved performance in terms of multiplexing, sensitivity, specificity, and dynamic range [20,21,22].

In this study, we developed a nucleic acid-based multiplex assay platform to easily and accurately detect multiple targets within 2 h using hydrogel microparticles, called barcode receptor-encoded particles (BREP). The BREP assay employs a set of BREPs and barcode probes, and the label-free barcode probe is target-specific; BREP is conjugated with a barcode receptor, which is not optimized for detecting an actual target but designed to hybridize with the barcode of the barcode probe. When the target is present, the barcode generated by the cleavage of the barcode probe is elongated after hybridization to the 3′-end of the barcode receptor of BREP, which leads to the production of a fluorescence signal due to the removal of the quencher from the reporter. Each BREP directly detects amplification products (barcodes) and not the target sequences, and it enables the identification of multiple targets through simple graphical encoding despite using a single fluorophore. To validate the feasibility of the BREP assay, we confirmed the generation of barcode and BREP fluorescence signals separately. To evaluate multiplexibility of the assay, we conducted 4-plex detection of the malaria species subtype. By changing the barcode sequence such that it is complementary to the target, the BREP assay can detect any genetic target, not only malaria species but also coronaviruses, which have many variants.

## 2. Materials and Methods

### 2.1. Materials

We purchased an SU-8 micro mold from MICROFIT (Hanam, Republic of Korea), PDMS (SYLGARD 184 Base & Curing Agent) from Dow Corning (Midland, MI, USA), 1.0 mm and 10.0 mm Miltex^®^ Biopsy Punch with Plunger from Integra Life Sciences (Princeton, NJ, USA), and microscope slides from Marienfeld Superior (Baden-Württemberg, Germany); polyethylene glycol diacrylate (PEGDA700; Mn = 700 Da), polyethylene glycol (PEG200; Mn = 200 Da), and a photoinitiator (Darocur 1173) were purchased from Sigma-Aldrich (St. Louis, MO, USA); streptavidin from Invitrogen (Waltham, MA, USA), 1× Phosphate-buffered saline (PBS) from Thermo Fisher Scientific (Waltham, MA, USA), TWEEN^®^20 from Sigma-Aldrich (St. Louis, MO, USA), and PCR master mix (QIAGEN Multiplex PCR Kit) and a nucleic acid extraction kit (QIAamp DNA Mini Kit) were purchased from Qiagen (Hilden, Germany). Taq polymerase, Taq buffer, dNTP (Dyne Taq Mg^2+^ plus), DNA staining solution (Dyne LoadingSTAR), and DNA size markers (Dyne 50 bp DNA Ladder) were purchased from Dyne Bio (Seongnam, Republic of Korea). BamH1 was purchased from New England Biolabs (Ipswich, MA, USA), 2× RNA loading dye from Thermo Fisher Scientific (Waltham, MA, USA), 1× TBE running buffer and 15% TBE-Urea Gel from Invitrogen (Waltham, MA, USA), and PAGEr^®^ Minigel Chamber from Lonza (Basel, Switzerland). All of Primer, Barcode probe, Barcode receptor, and target DNA template used for the test were synthesized from Bioneer (Daejeon, Republic of Korea) Sequences were described in Appendix A.

For developing this assay, we used an inverted microscope (Axiovert 200; Zeiss, Jena, Germany), a pressure regulator (ITV0031-3BL; SMC, Tokyo, Japan), an LED lamp (M365LP1; Thorlabs, Newton, NJ, USA), a UV power meter (GT-512; Gilwoo Trading Co., Seoul, Republic of Korea), a photomask (50,000 dpi) printed from HAN&ALL Tech (Ansan, Republic of Korea), a thermal shaker (TMS-300; ALLSHENG, Hangzhou, China), and a thermal cycler (S1000; BIO-RAD, Hercules, CA, USA).

### 2.2. BREP Assay Design

As shown in Figure 1a, the two important components of the BREP assay are the ”BREP” (with barcode receptor) and the “barcode probe.” The BREP consists of hydrogel particles functionalized with streptavidin and a biotinylated barcode receptor. The barcode receptor is dually labeled with a reporter located upstream from the restriction region and a quencher controlling the reporter; it is composed of a c-barcode region complementary to the barcode region of the barcode probe and an extension region including a restriction region recognized by a restriction enzyme. The barcode probe is a single-stranded oligonucleotide composed of a barcode region present adjacent to the probe region, which is complementary to the target region of interest. The probe region of the barcode probe is designed to have similar specificity to primers. In addition, G-C contents were adjusted so that the annealing temperature of the target region was similar to that of the primers. The barcode region of the barcode probe is a unique sequence that does not hybridize with a target sequence or any off-target materials such as primers. It is advantageous to increase the length of the sequence to prevent the barcode region from hybridizing with the target, but this is disadvantageous for binding to the barcode receptor immobilized on the hydrogel surface. Therefore, we designed 15 nt, which is slightly shorter than typical primer designs.

The BREP assay comprises the following three steps:(1)Amplification: generation of barcode

When the target DNA is present, the probe region of the barcode probe anneals to the specific sequence of the target region, the upstream primer is elongated by polymerase and the barcode is released.

(2)Reaction 1: hybridization and extension

The barcode released from the barcode probe hybridizes to the c-barcode region of the barcode receptor, and the barcode extends to the restriction region to form an extended double-strand.

(3)Reaction 2: restriction

Duplex barcode receptor causes cleavage of the restriction site by restriction enzyme and the quencher is removed at the same time; finally, the reporter generates a fluorescence signal.

### 2.3. Hydrogel Particle Synthesis

Hydrogel particles were synthesized as per established method (Hyunjee Lee et al., 2013), with some modifications [23]. The prepolymer solution used for particle synthesis consisted of 20% (*v*/*v*) polyethylene glycol diacrylate (PEGDA700; Mn = 700 Da), 40% (*v*/*v*) polyethylene glycol (PEG600; Mn = 200 Da), 35% (*v*/*v*) deionized (DI) water, and 5% (*v*/*v*) Darocur 1173 (photoinitiator). The prepolymer solution was vortexed for 1 min and loaded onto a microchannel (height: 50–38). One cycle of particle synthesis via stop-flow lithography consisted of 400 ms of flow, 200 ms of stop, 60 ms of UV exposure at 2000 $\mathrm{mW}\cdot {\mathrm{cm}}^{-1}$, and 140 ms of hold time. After synthesis, the particles were collected in a microtube and washed three times using 1× PBST. For particle functionalization, 50 µL of particles (100 particles ${µL}^{-1}$ were incubated with 50 µL of 5 $\mathrm{mg}\cdot {\;\mathrm{mL}}^{-1}$ streptavidin and 1.2 µL of 0.5 M NaOH for 24 h at 1500 rpm at 25 °C. After functionalization, the particles were washed three times with 1× PBST.

### 2.4. Barcode Receptor Immobilization on Streptavidin Functionalized Hydrogel Particles

BREPs were synthesized as follows: 10 μL of functionalized particles (500 particles ${µL}^{-1}$) were incubated with 10 μL of 20 µM barcode receptor for 2 h at 2000 rpm at 25 °C. The BREPs were washed thrice with 1× PBST and stored at 4 °C.

### 2.5. BREP Assay

The amplification step was performed in the thermal cycler. The reaction mixture consisted of 2 μL of DNA template, 1 μL each of 10 μM forward primer and reverse primer, 0.5 μL of 10 pmol/μL barcode probe, 12.5 μL of the PCR master mix, and 8 μL of nuclease-free water in a final volume of 25 μL. The amplification conditions were as follows: denaturation at 95 °C for 15 s; annealing and extension at 60 °C, 1 min, 1 min 30 s, 2 min for 39 cycles after the pre-denaturation step at 95 °C for 3 min.

Reactions 1 and 2 were performed at 2000 rpm for 20 min at 45 °C in the thermal shaker. The amplicons were mixed with 2.5 μL of 10× Taq buffer, 10 μL of BREP (approximately 20 ea), 2 μL of Taq, 2 μL of 2 mM dNTP, 2 μL of BamH1, 2.5 μL of BamH1 buffer, and 28.5 μL of nuclease free water in a final volume of 50 μL.

### 2.6. Validation of Barcode Generation

The generated barcode was validated using Urea Gel PAGE, and barcode generation was indirectly confirmed as the amplification step worked as per 2% agarose gel electrophoresis. After the amplification step, for PAGE, 5 μL of the PCR product was mixed with 5 μL of 2× RNA loading dye and loaded onto 15% TBE-Urea Gel. The gel was run at 150 V for 50 min. For agarose electrophoresis, 5 μL of PCR products was mixed with 1 μL of 6× Dyne LoadingSTAR and loaded onto a 2% agarose gel. The gel was run at 100 V for 35 min.

### 2.7. Multiplexed Detection of Malarial Species

PCR was performed in the thermal cycler using the following reaction mixture: 2 μL of the DNA template, 1 μL each of forward and reverse primers (10 μM; total 4 μL), 0.5 μL each of the barcode probes (10 μM; total 2 μL), 12.5 μL of the PCR master mix, and 4.5 μL of nuclease free water, with the final volume being 25 μL. The temperature conditions were as follows: denaturation at 95 °C for 15 s; annealing and extension at 60 °C, 1 min 30 s for 39 cycles after the pre-denaturation step at 95 °C for 3 min. For barcode hybridization and extension, reactions 1 and 2 were performed for 20 min at 45 °C and 2000 rpm in the thermal shaker; 20 μL of PCR product was mixed with 2.5 μL of 10× Taq buffer, 10 μL of BREP (approximately 20 ea), 2 μL of Taq polymerase, 2 μL of 2 mM dNTP, 2 μL of BamH1, 2.5 μL of BamH1 buffer, and 28.5 μL of nuclease free water in a final volume of 50 μL.

Clinical samples of malaria were collected from 2014 to 2016 at the Korea University Guro Hospital, Republic of Korea. Malaria species was confirmed in all samples by microscopic examination. Of the four malarial parasites, the target subtype was identified as *P. falciparum*. The parasites were cultured and DNA extraction was performed using a Qiagen nucleic acid extraction kit.

### 2.8. Analysis of Fluorescence Intensity of BREP

Fluorescence images of BREP were captured using an Auto F-scanner (JULI FL) with an exposure time of 30 ms. The brightness of the fluorescence signals was measured using an image processing program (ImageJ, Version 1.46r).

## 3. Results

### 3.1. Synthesis of BREP

BREP was synthesized with polyethylene glycol (PEG), using stop-flow lithography. We used a hydrogel because it provides a solution-like environment, which enhances the loading capacity. Moreover, the PEG matrix easily conjugates with the desired functional groups [19,23]. During stop-flow lithography, incomplete free-radical polymerization reactions create unreacted sites in the hydrogel network that are still active as acrylate groups [24]. In the present study, these unreacted active sites were used for streptavidin conjugation. BREPs are universally applicable because they utilize the streptavidin-biotin interaction, which is widely used in biotechnology due to its high affinity and stability [25]; thus, the BREP assay is not limited to nucleic acids but can be used for all types of biotinylated materials.

Although spectrometric encoding with fluorophores is widely used in various assay platforms, it causes spectral overlap; however, graphically encoded BREPs, with combinations of location and number of holes on particles (Figure 1b) exhibit a higher encoding capacity, which can be easily achieved by changing the location and the number of holes [26].

### 3.2. BREP Assay Optimization

Several factors should be considered to ensure optimum performance of an assay; optimization leads to higher molecular sensitivity, specificity, and precision. Therefore, we optimized the conditions for the BREP assay. We analyzed the effects of time and temperature on the results of the assay. To determine the optimum reaction time, we performed the assay at different times, but a significant difference was not observed in the fluorescence signal of the positive sample among the three different durations investigated. However, we observed a slight increase in the fluorescence signal in the positive sample, as the time increased, but at the same time, the fluorescence signal in the negative and blank samples increased, indicating that non-specific binding increased with time (Figure 2a). Because an assay is more convenient if it can be performed in a short period, the optimum time for the BREP assay was considered to be 20 min, during which it exhibited a high fluorescence signal in positive samples and low non-specific binding. To explore the effect of temperature, we performed the assay at different temperatures. As shown in Figure 2b, a lower fluorescence signal was observed in the negative and blank samples as the temperature increased from 37 °C to 50 °C, implying that non-specific binding was reduced. However, when the temperature increased above 45 °C, fluorescence signals in the positive samples also decreased. Therefore, high reaction temperature not only increased the specificity but also prevented binding. Consequently, the optimum temperature of the assay was determined to be 45 °C, owing to the fluorescence signals observed in the positive samples.

### 3.3. Validation of BREP Assay

To validate the feasibility of the BREP assay, we independently verified whether the amplification step (barcode generation) and reactions 1 and 2 (hybridization, extension, and restriction) produced accurate results. Fluorescence signal generation is directly dependent on the extension of the barcode on the barcode receptor and not on template amplification. To validate the generation of the barcode (single-stranded nucleotides) in the amplification step, we performed urea PAGE electrophoresis for comparison with the synthetic barcode (16 nt). A 16-nt amplicon was generated only in the presence of a positive DNA template, as shown in Figure 3a. The barcode can also be amplified according to the amplification cycle number, which leads to high sensitivity.

We validated reactions 1 and 2 by creating a standard curve for the synthetic barcode with different concentrations but not with the amplicons generated in the amplification step. The barcode successfully hybridized with the barcode receptor and the barcode was elongated. Finally, the quencher was cleaved by the restriction enzyme, and a dose-dependent fluorescence signal was detected (Figure 3b).

Thus, the sensitivity of the BREP assay was verified. Next, we investigated the limit of detection (LOD) of the assay by varying the concentration of the synthetic DNA template. In the experiment, barcode generation was indirectly confirmed by agarose electrophoresis, which confirmed that the amplification step worked, and a subsequent process was performed. The results showed that as the concentration of DNA template was increased, more barcodes were generated during the amplification step (Appendix A). We observed that the final fluorescence signal of BREP was proportional to the amplification of the DNA template but was independent of the DNA template. The lowest detectable concentration was 1.53 fM, resulting in the detection ability over a wide range of 4 log (Figure 3c).

### 3.4. Multiplexed Detection of Malaria Species

Next, we verified the specificity of the BREP assay in detecting multiple targets simultaneously. In the BREP assay, due to graphical encoding, multiple BREPs paired with barcode probes can effectively distinguish multiple targets in the same well. In this study, we targeted four species of the malarial parasite for multiplexing (*P. falciparum*, *P. vivax*, *P. ovale*, and *P. malariae*). Malaria, one of the most prevalent parasitic diseases, poses a threat to human health worldwide, especially for children. The World Health Organization (WHO) estimated that 241 million cases of malaria and 627,000 deaths from malaria occurred globally in 2020 [27]. Therefore, rapid and accurate diagnosis of malaria is important because misdiagnosis leads to inappropriate treatment, which not only delays complete recovery, but also results in overdose of drugs and spread of anti-malaria drug resistance [28,29]. Furthermore, the identification of malarial parasite species is critical for appropriate treatment [30,31,32]. Efforts are being made to develop practical diagnosis methods to control malaria, such as microscopy-based, rapid diagnostic tests (RDTs), and techniques to detect nucleic acids specific to each species that are highly sensitive and specific [33,34,35].

In the present study, the four types of barcodes of BREPs in a single well determined whether the patient was infected by the malarial parasite and identified the species of the parasite. Synthetic DNA templates of different species of the malarial parasite were analyzed to verify their performance. Each target DNA template was mixed with four sets of BREPs and barcode probes. Likewise, before the BREP assay, we confirmed barcode generation in the amplification step with four types of barcode probes. As shown in Figure 4a, four types of barcodes were perfectly generated, and the fluorescence signal of each BREP was detected only in response to the presence of the target DNA template for each species with low cross-reactivity (Figure 4b). This is due to the use of a set of BREP and barcode probes that were carefully designed to generate a target-independent signal; they can be applied to other genetic targets by changing only the probe region of the barcode probe. Furthermore, the intensity of the fluorescence signal for the 4-plex assay was comparable to that of any singleplex assay, demonstrating that the BREP assay is compatible with multiplexed assays (Figure 4c). In this study, we obtained a fluorescence signal similar to singleplex, but using BREP technology for singleplex or analysis of a small number of targets can be wasteful. This is because designing BREP requires an additional step to produce particles with barcode probes and DNA immobilized. However, the method that can confirm the detection result according to the shape of the particle, which is the technology we have proposed, becomes superior to the existing PCR technology as the number of targets to be detected increases. In addition, since multiple analysis is possible within one PCR tube, the total amount of samples used is reduced.

Moreover, singleplex and multiplex BREP assays were performed with clinical samples from patients with malaria (subtype = *Plasmodium falciparum* (P.f)). Four different clinical samples were analyzed independently in a single reaction, and the results were compared with those of the multiplex samples. Firstly, we confirmed barcode generation in the amplification step with clinical samples from four patients infected with P.f (Appendix A). As shown in Figure 5a, we identified an infection. Moreover, no significant differences in the fluorescence signal intensity were observed between the singleplex and multiplex assays, indicating that the BREP assay maintains intact sensitivity even in the presence of multiple analytes. When the experiment was performed using a synthetic DNA template, the results, without any differences, exhibited the potential applicability of the BREP assay in clinical analyses (Figure 5b).

In the present study, we performed multiplex detection targeting DNA. However, our target is not limited to DNA. We utilize PCR for the initial step of the reaction. By replacing the PCR process with the RT-PCR process, multiple detection of RNA is also possible. This is because barcode DNA can be generated even in the RT-PCR process. Therefore, it can be said that this is a powerful genetic analysis technology because it can perform various multi-analyses even for RNA targets such as viruses.

## 4. Conclusions

Due to the growing importance of nucleic acid analysis, accurate and cost-effective multiplexing of nucleic acids is essential. The BREP assay is a multiplexed hydrogel particle-based nucleic acid analysis assay that employs a unique pairing of BREP with barcode receptors and barcode probes.

The combination of BREP and barcode probes is flexible and applicable to any nucleic acid targets via designing of a set of barcode receptors and barcodes. In addition to its flexibility, the assay is cost-effective due to the combination of BREP and barcode probe; the barcode receptor of BREP is independent of the target sequences and can be synthesized on a large scale. Barcode probes specific to the target sequence do not require labels and can be synthesized at a lower price than the probes with labels.

Multiple barcode receptors conjugated with graphically encoded hydrogel particles can be detected in a single reaction using the same fluorophore. In this study, the multiplex BREP assay was carried out using four different encoding particles in a single well, which took less than 1 h to identify four species of the malarial parasite.

Thus, by introducing high-capacity hydrogel particles and sets of barcode receptors and probes, the workflow was simplified, which reduced the detection time (<2 h) and hands-on time, making the assay user-friendly.

## Figures and Tables

**Figure 1 biomedicines-10-03246-f001:**
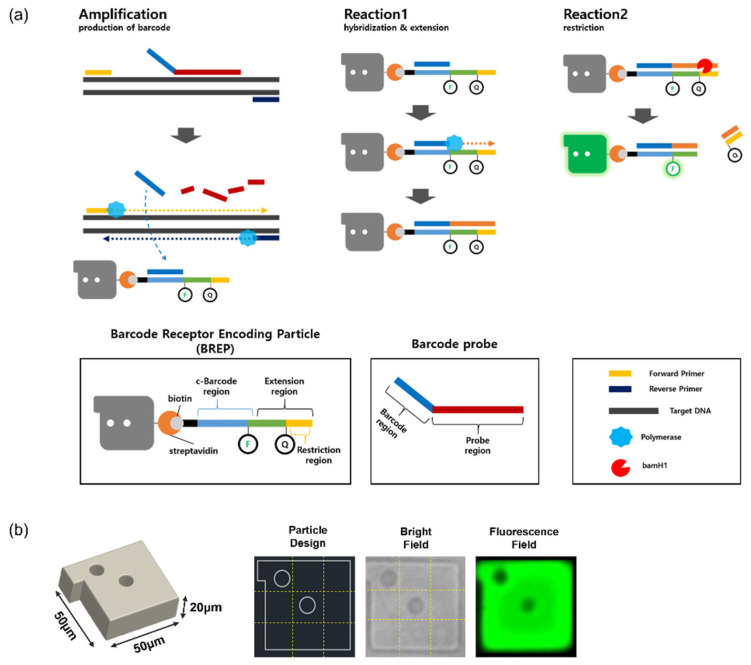
(**a**) Schematic illustration of Barcode Receptor Encoding Particle (BREP) assay. (**b**) Particle image and graphical encoding created with combinations of holes on square shape particles with a protruding bar.

**Figure 2 biomedicines-10-03246-f002:**
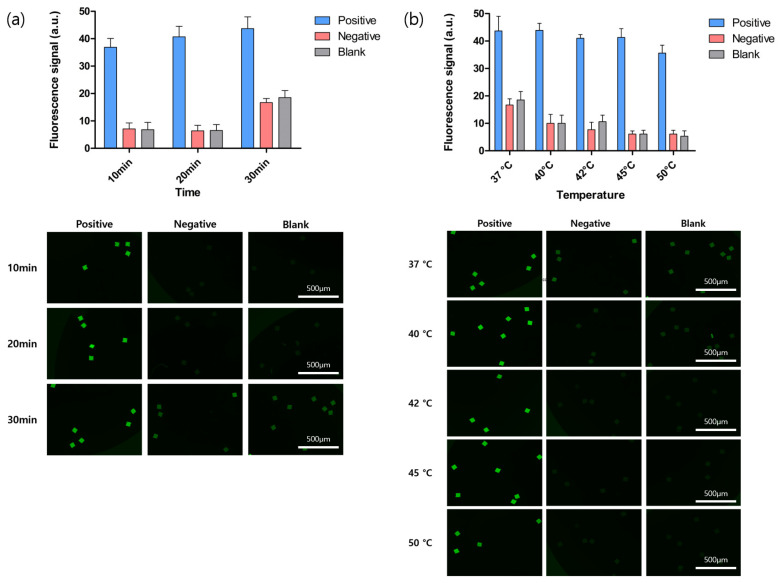
Optimization of BREP assay conditions. (**a**) Optimization of assay duration conducted at 45 °C with 10 pmol/mL of synthetic barcode for positive sample, 10 pmol/mL of barcode probe for negative sample, and deionized water as blank. Fluorescence signal intensity at 10, 20, and 30 min. (**b**) Temperature optimization was conducted with 10 pmol/mL of synthetic barcode for positive sample, 10 pmol/mL of barcode probe for negative sample, and deionized water as blank, with the assay duration being 20 min. Fluorescence signal intensity at 37, 40, 42, 45, and 50 °C. Bar graph showing mean values of fluorescence signal intensity. Error bars represent standard deviation (n = 3).

**Figure 3 biomedicines-10-03246-f003:**
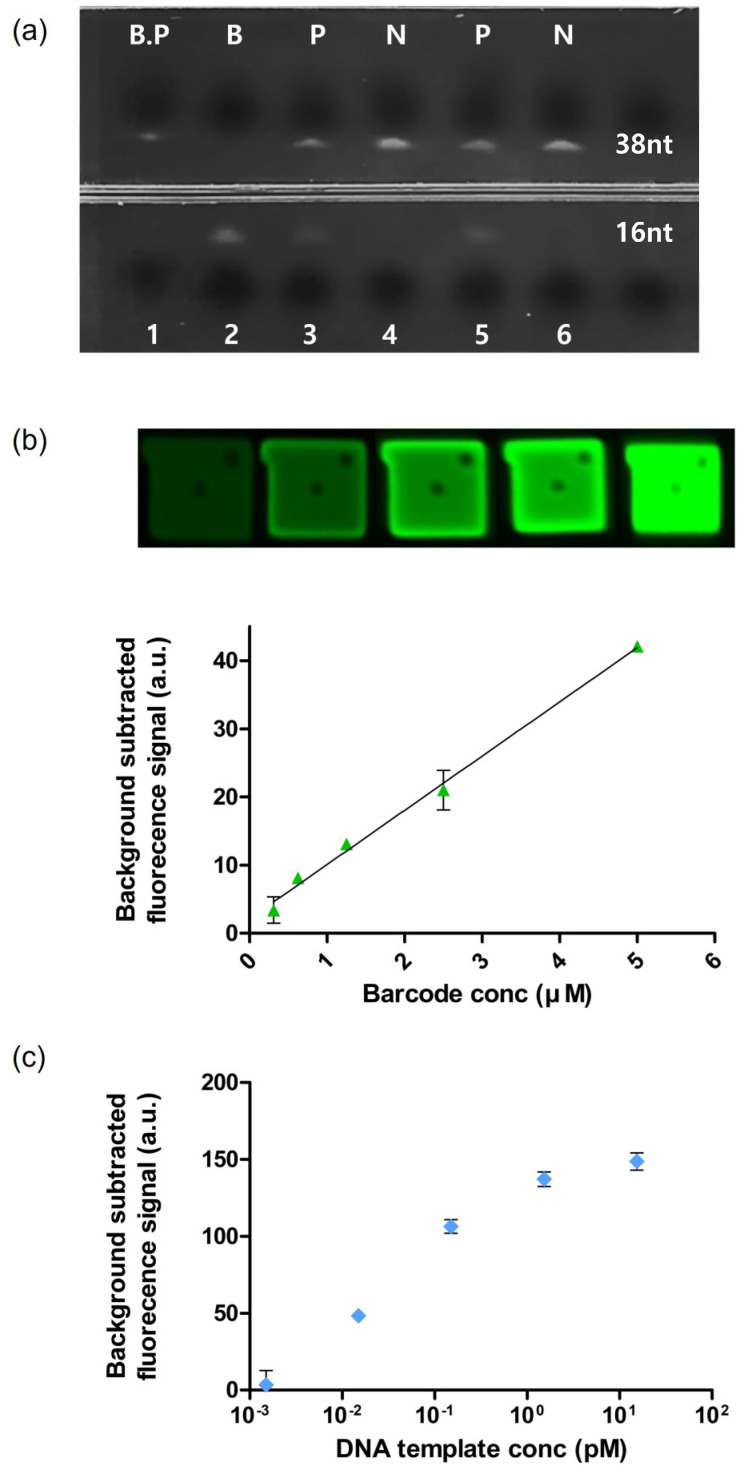
(**a**) PAGE to detect the size of the amplicon (16 nt). Lane 1: Barcode probe (38 nt); Lane 2: Barcode (16 nt); Lanes 3 and 5: amplicons of positive DNA template; Lanes 4 and 6: amplicons of negative DNA template. (**b**) Standard curve fitted by fluorescence signal intensity with respect to barcode concentrations of 0.3125, 0.625, 1.25, 2.5, 5, and 10 μM (standard deviation (SD) of each value was 1.93, 0.39, 0.13, 2.92, 0.20, and 2.73, respectively; n = 3) (**c**) Fluorescence signal intensity with respect to DNA template concentrations of 15.38, 1.53, 0.15, 0.015, and 0.0015 pM (standard deviation (SD) of each value was 4.51, 3.84, 3.61, 1.62, 7.68, 1.00, and 0.77, respectively; n = 3).

**Figure 4 biomedicines-10-03246-f004:**
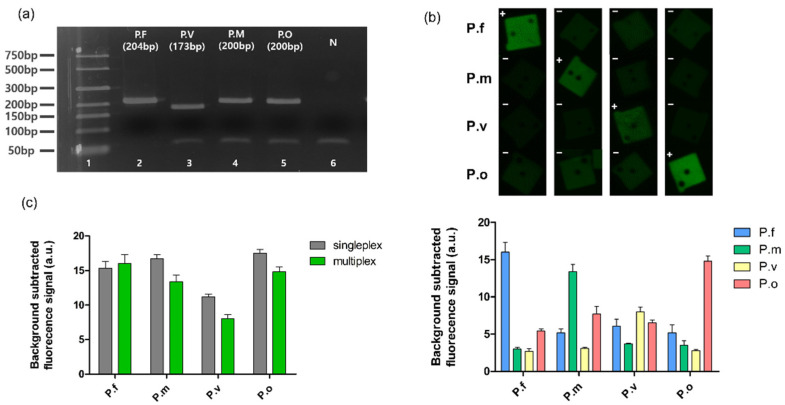
Multiplex BREP assay. (**a**) Validation of amplification using agarose gel electrophoresis (Multiplex) Lane 1: 50 bp DNA ladder; Lanes 2–5: amplification step product with *Plasmodium falciparum* (P.f), *P. vivax* (P.v), *P. malariae* (P.m), and *P. ovale* (P.o) of 0.2 ng/μL DNA template; Lane 6: negative control, deionized water. (**b**) 4-plex assay to detect four species of malarial parasite. The + and – signs indicate the presence (+) and absence (−) of the target DNA template. (**c**) Comparison of the results obtained after singleplex and multiplex assays. Bar graph showing mean values of background subtracted from the fluorescence signal intensity. Error bars represent standard deviation (n = 3).

**Figure 5 biomedicines-10-03246-f005:**
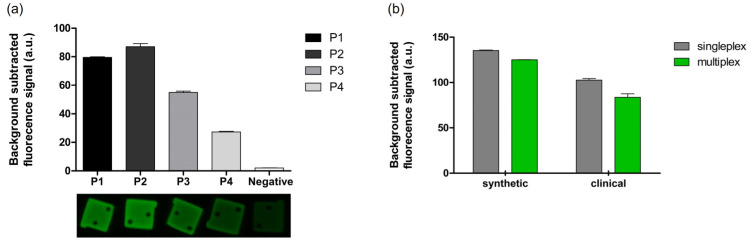
BREP assay with clinical samples. (**a**) Singleplex BREP assay with clinical sample of four patients with malaria. (**b**) Comparison of the results obtained after singleplex and multiplex assays with both synthetic DNA template and clinical samples from patients with malaria. Bar graph showing mean values of background subtracted from fluorescence signal intensity. Error bars represent standard deviation (n = 3).

## Data Availability

The datasets used in this study are available from the corresponding author upon reasonable request.

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
