# Peer review of "Multiplex Assay for Rapid Detection and Analysis of Nucleic Acid Using Barcode Receptor Encoded Particle (BREP)"

_biomedicines, 2022, doi:10.3390/biomedicines10123246_

Round 1
Reviewer 1 Report
In this manuscript, the authors report hydrogel microparticle-based nucleic acid multiplex platform named Barcode Receptor Encoded Particle. The presented methods are appropriate and results are convincing. The presented method has a potential to contribute and improve the multiplex assay. The manuscript is well prepared.
Author Response
Dear Reviewers.
Thanks for reviewers noble comments and valuable reviews.
Based on the reviewer's opinion, the manuscript was amended as follows.
All modifications are highlighted in the updated manuscript
Reviewer1: Are all the cited references relevant to the research? Not Applicable
We double-checked the reference article to confirm its relevance.
English language and style are fine/minor spell check required
English proofreading was completed for the entire manuscript.

Reviewer 2 Report
I recommend this manuscript for the publication
Author Response
Dear Reviewers.
Thanks for reviewers noble comments and valuable reviews.
Based on the reviewer's opinion, the manuscript was amended as follows.
All modifications are highlighted in the updated manuscript
Reviewer2: Extensive editing of English language and style required
English proofreading was completed for the entire manuscript.

Reviewer 3 Report
The manuscript describes a multiplex assay for detection of DNA sequences. The authors said that the method can be easily multiplexed and applied to others DNA sequences.
It seems to be useful and of broad interest, but some aspects need to be clarified or better explained.
The authors need to explain better the method, including for example the limitations diue to the length of the sequences, probes, and barcode regions, the limitations depending on the sequence of the fragment, and so on.
Because the aim is to describe an open method to diagnose the presence of DNA sequences, the authors should estimate the cost of the method, in terms of money bur also in time to develop a new protocol.
Any restriction to the technique must be also explained in the article, weakness in the protocol must also be declared. I imagine the protocol could be also used for RNA detection. Please mention the implications for that analysis.
Figure 1a must be improved, including the two DNA strands and both primers, forward and reverse.
Supplementary Table 1, sequences must be checked, I think there is a sequence error at least in Barcode region of the Barcode probe or the C-barcode region complementary to the barcode region in P.f.
An effort can be done to improve the language used in the manuscript.
Author Response
Dear Reviewers.
Thanks for reviewers noble comments and valuable reviews.
Based on the reviewer's opinion, the manuscript was amended as follows.
All modifications are highlighted in the updated manuscript
Reviewer3: The manuscript describes a multiplex assay for detection of DNA sequences. The authors said that the method can be easily multiplexed and applied to others DNA sequences.
It seems to be useful and of broad interest, but some aspects need to be clarified or better explained.
We appreciate the reviewer's noble comments. Based on your suggestion, we have added the following to page 1, paragraph 1.
The amplification of multiple nucleic acids enables the acquisition of diagnostic information, e.g., identification of infectious agents, genetic variations, mutant genes, or single nucleotide polymorphism (SNP) related to particular diseases, even though the amount of sample that can be acquired is limited or the concentration of analytes in the sample is low-level, which is not possible in case of single-target serial testing.
The authors need to explain better the method, including for example the limitations due to the length of the sequences, probes, and barcode regions, the limitations depending on the sequence of the fragment, and so on.
We appreciate the reviewer's noble comments. Based on your suggestion, we have added the following to page 3, paragraph 3.
"The probe region of the barcode probe is designed to have similar specificity to primers. In addition, G-C content were adjusted so that the annealing temperature of the target region was similar to that of the primers. The barcode region of the barcode probe is a unique sequence that it does not hybridize with target sequence or any off-target materials like primers. It is advantageous to increase the length of the sequence to prevent the barcode region from hybridizing with the target, but it is disadvantageous for binding to the barcode receptor immobilized on the hydrogel surface. Therefore, we designed 15 nt, which is slightly shorter than typical primer designs."
Because the aim is to describe an open method to diagnose the presence of DNA sequences, the authors should estimate the cost of the method, in terms of money but also in time to develop a new protocol.
You asked a very important question.
The technology we developed began with a new challenge for multiple detection of DNA sequences. After devising a new method, discovering an appropriate enzyme, and then determining the optimal reaction temperature, reaction time, and barcode sequence, it took many iterations. In addition, it took a lot of time to conduct verification experiments at each stage. It took about two years from the conception stage to confirming the results of clinical samples. However, it is expected that much less time and effort will be required to detect new targets in the future due to the knowledge accumulated in this process. In addition, we believe that if a gene analysis company, rather than a university laboratory like ours, applies these technologies, it will be possible to develop them with much less time and cost.
Any restriction to the technique must be also explained in the article, weakness in the protocol must also be declared. I imagine the protocol could be also used for RNA detection. Please mention the implications for that analysis.
Based on the reviewer's opinion, the following contents were added to the manuscript.
Page 9, paragraph 2.
“In this study, we obtained a fluorescence signal similar to singleplex, but using BREP technology for singleplex or small number of target analysis can be wasteful. This is because designing BREP requires an additional step to produce particles with barcode probes and DNA immobilized. However, the method that can confirm the detection result according to the shape of the particle, which is the technology we proposed, becomes superior to the existing PCR technology as the number of targets to be detected increases. In addition, since multiple analysis is possible within one PCR tube, the total amount of samples used is reduced.”
Page 9, paragraph 4.
“In the present study, we performed multiplex detection targeting DNA. However, our target is not limited to DNA. Because we utilize PCR for the initial step of the reaction. By replacing the PCR process with the RT-PCR process, multiple detection of RNA is also possible. This is because barcode DNA can be generated even in the RT-PCR process. Therefore, it can be said that it is a powerful genetic analysis technology because it can perform various multi-analyses even for RNA targets such as viruses.”
Figure 1a must be improved, including the two DNA strands and both primers, forward and reverse.
As pointed out in Figure 1a, both forward and reverse primers were included.
Supplementary Table 1, sequences must be checked, I think there is a sequence error at least in Barcode region of the Barcode probe or the C-barcode region complementary to the barcode region in P.f.
Thank you for your careful review.
We found an error in the DNA sequence presented and corrected Supplementary Table 1.
(5'-[BHQ-1] AGGACGGGATCCACA[FAM(dT)]TGAATCGCACTATTCTTTTTTTTT[biotin]-3')
An effort can be done to improve the language used in the manuscript.
In order to improve the language of the manuscript, English proofreading was completed for the entire manuscript.
